# Eosinophil Granule Proteins Involvement in Acute Appendicitis—An Allergic Disease?

**DOI:** 10.3390/ijms24109091

**Published:** 2023-05-22

**Authors:** Nuno Carvalho, Elisabete Carolino, Hélder Coelho, Ana Lúcia Barreira, Luísa Moreira, Margarida André, Susana Henriques, Carlos Cardoso, Luis Moita, Paulo Matos Costa

**Affiliations:** 1Serviço Cirurgia Geral, Hospital Garcia de Orta, 2805-267 Almada, Portugal; analuciapb19@gmail.com (A.L.B.); susanahenriques@campus.ul.pt (S.H.); paulomatoscosta@gmail.com (P.M.C.); 2Faculdade Medicina, Universidade Lisboa, 1649-028 Lisboa, Portugal; 3H&TRC—Health & Technology Research Center, ESTeSL—Escola Superior de Tecnologia da Saúde, Instituto Politécnico de Lisboa, 1549-020 Lisboa, Portugal; etcarolino@estesl.ipl.pt; 4Serviço de Anatomia Patológica, Hospital Garcia de Orta, 2805-267 Almada, Portugal; hmoc85@gmail.com; 5Serviço de Urologia, Hospital Garcia de Orta, 2805-267 Almada, Portugal; aluisa.mam@hotmail.com (L.M.); margarida_andre@hotmail.com (M.A.); 6Dr. Joaquim Chaves Laboratório de Análises Clínicas, 1495-068 Alges, Portugal; carlos.cardoso@jcs.pt; 7Innate Immunity and Inflammation Lab, Instituto Gulbenkian de Ciência Oeiras, 2780-156 Oeiras, Portugal; lmoita@igc.gulbenkian.pt; 8Instituto de Histologia e Biologia do Desenvolvimento, Faculdade Medicina, Universidade Lisboa, 1649-028 Lisboa, Portugal

**Keywords:** appendicitis, allergy, eosinophils, Eosinophil-derived neurotoxin, eosinophil cation protein, eosinophil peroxidase, hypersensitivity type 1 reaction

## Abstract

Several pieces of evidence point to an allergic component as a trigger of acute appendicitis. As the Th2 immune response is characterized by eosinophil mobilization to the target organ and release of their cationic granule proteins, it is reasonable to investigate if the degranulation of eosinophils could be associated with the local injury. The primary aim of this study is to evaluate the participation of eosinophils granules proteins in acute appendicitis, both at local and systemic levels and the secondary aim is to evaluate the diagnostic accuracy of eosinophils granules proteins for the detection of acute appendicitis, as well as for distinguishing between complicated and uncomplicated acute appendicitis. Eosinophil-derived neurotoxin (EDN), eosinophil cationic protein (ECP) and eosinophil peroxidase (EP) are the most well-known eosinophil granule proteins. From August 2021 to April 2022, we present a prospective single-center study to evaluate the EDN, ECP, and EP concentrations simultaneously in appendicular lavage fluid (ALF) and the serum of 22 patients with acute phlegmonous appendicitis (APA), 24 with acute gangrenous appendicitis (AGA), and 14 normal controls. Concerning EDN, no differences were found between groups. ECP concentrations in ALF and serum were significantly higher in the histologically confirmed acute appendicitis compared to the control groups (*p* < 0.0001 and *p* < 0.0001, respectively). In ALF, no differences were found between ECP levels in APA: 38.85 ng/mL (IQR 26.50–51.77) and AGA 51.55 ng/mL (IQR 39.55–70.09) groups (*p* = 0.176). In the serum, no difference was found between ECP levels at APA: 39 ng/mL (IQR 21.30–56.90) and AGA: 51.30 ng/mL (IQR 20.25–62.59) (*p* = 0.100). For EP, the concentrations in ALF (*p* < 0.001) and serum (*p* < 0.001) were both higher in acute appendicitis compared to the control. In ALF, no difference was found between APA: 240.28 ng/mL (IQR 191.2–341.3) and AGA: 302.5 (IQR 227.7–535.85) (*p* = 0.236). In the serum, no differences were found between APA: 158.4 ng/mL (IQR 111.09–222.1) and AGA: 235.27 (IQR 192.33–262.51) (*p* = 0.179). Globally, the ALF concentrations were higher than serum concentrations, reflecting an intense inflammatory local reaction in AA. The optimal ECP cut-off for discriminating between acute appendicitis and the controls was >11.41 ng/mL, with a sensitivity of 93.5%, but with a specificity for identifying appendicitis of 21.4%, good discriminative power (AUC = 0.880). For EP, the optimal cut-off was >93.20 ng/mL, with a sensitivity of 87%, but with a specificity of 14.3% (AUC = 0.901), excellent discriminative power. For the diagnosis of perforated AA, the discriminative power of ECP and EP serum concentrations are weak (AUC = 0.562 and AUC = 0.664, respectively). Concerning the presence of peritonitis, the discriminative power of ECP and EP serum concentrations is acceptable, respectively: AUC = 0.724 and AUC = 0.735. Serum levels of EDN (*p* = 0.119), ECP (*p* = 0.586) and EP (*p* = 0.08) in complicated appendicitis were similar to uncomplicated appendicitis. Serum concentrations of ECP and EP can be added to decision-making AA diagnosis. A Th2-type immune response is present in AA. These data bring forward the role of an allergic reaction in the pathogenesis of acute appendicitis.

## 1. Introduction

Acute Appendicitis (AA) remains one of the most common reasons for emergency surgery [1]. Its etiology is still not fully understood [2]. Recent evidence has suggested that AA has an allergic component [3,4,5].

The intestine has most kinds of immune cells (e.g., T cells, B cells, mastocytes, and macrophages) and it is continually exposed to foreign antigens, such as dietary components, pollutants, harmful pathogens, and commensal bacteria. To maintain intestinal homeostasis, mucosal immunity must distinguish between friends and foes [6].

Eosinophils are innate immune granulocytes involved in the defense against parasites and in the pathogenesis of Th2 immune-mediated disorders [7]. Eosinophils modulate allergic inflammation [8] and eosinophilia has been associated with uncomplicated appendicitis in children [9,10].

Eosinophil granules contain a considerable variety of pre-formed biologically active substances, such as cytotoxic cationic proteins, ready for rapid release [7]. Eosinophil-derived neurotoxin (EDN), eosinophil cation protein (ECP), or eosinophil peroxidase (EP) are among the most studied eosinophil proteins, namely in asthma [11,12]. 

ECP, EP, and EDN granule proteins, released by eosinophils, are markers of eosinophil activation and have been identified as potential biomarkers of type 2 eosinophilic disease [13].

Eosinophil-derived cytotoxic mediators have a main role in tissue damage, as documented in Th2 asthma and other eosinophilic-associated inflammatory conditions [14,15].

Eosinophil muscle infiltration is common in AA [16,17]. If eosinophils are not simple bystanders, there will be degranulation, with consequent elevation of protein granules concentration, both at local and systemic levels.

The clinical diagnosis of AA is challenging because many symptoms may be nonspecific and the presentation can be variable [18].

Despite the radiological and analytical advances of the last decades, there is still a significant rate of diagnostic error in AA [19]. Most routinely disposable inflammatory biomarkers are not sensitive or specific enough to confirm or exclude an AA diagnosis [20].

Although appendectomy is the gold standard of treatment for AA, recent studies indicate that uncomplicated AA may also be treated conservatively [21]. It is of paramount importance to distinguish between complicated and uncomplicated AA in the decision-making process regarding the best treatment modality, conservative vs. operative treatment [22].

In addition to standard biomarkers several new biomarkers for AA such as hyponatremia [18,23], hyperfibrinogenemia [23], hyperbilirubinemia [22], leucine-rich α-2-glycoprotein 1 [21], neutrophil gelatinase-associated lipocalin (NGAL) [24], or interleukin-6 (IL-6) [19,24] have been recently investigated and presented good predictive values for the detection of acute appendicitis, as well as for distinguishing between complicated and simple AA. A more recent study showed that non-invasive markers from saliva, such as leucine-rich α-2-glycoprotein 1 (LRG1) may be a promising biomarker for the detection of acute appendicitis in children [20].

The primary purpose of this study was to investigate the putative role of eosinophil granule proteins in AA, both at a systemic level, in peripheral blood, and at the local level, in the appendix. For evaluating the local level eosinophilic role, we developed an innovative methodology, the appendicular lavage fluid (ALF), that reflects the local immunoinflammatory milieu in the appendix [25]. 

The second purpose was to evaluate the discriminatory power of eosinophils proteins concentrations in the diagnosis of AA and distinguish complicated from uncomplicated appendicitis.

If AA is an allergic reaction, it will be expected that eosinophilic proteins are elevated in AA, both at systemic and local levels, and can help in the diagnosis of AA and distinguish complicated from uncomplicated appendicitis.

## 2. Results

### 2.1. Patients

The main demographic characteristics of the enrolled group of 46 patients and the 14 control patients were reflected in Table 1.

All the patients with the clinical diagnosis of AA that were submitted to appendectomy met the histologic criteria of AA. No helminth or parasite was found in the slices. 

There was a significant statistical difference between groups, regarding age, which was maintained during pair-wise analysis for the pair Control-AGA, with age being higher in the control group (Table 1). Regarding sex, BMI, and history of allergy, no differences were found. (Table 1).

### 2.2. Hemogram and Appendicular Histology

Hemogram and appendicular histology are depicted in Table 2.

Regarding WBC, differences were found between groups (*p* < 0.001). Through multiple paired comparisons, the differences found were between the control group and APA (*p* < 0.001) and the control group and AGA (*p* < 0.001) group, with the highest levels being related to APA and AGA groups.

Concerning neutrophils, differences were found between groups (*p* = 0.005), and at multiple paired comparisons, the difference was for the pair AGA-Control (*p* = 0.004), having AGA the higher levels.

Respecting monocytes, differences were found between groups (*p* = 0.035), and at multiple paired comparisons, the difference was also for the pair AGA-Control (*p* = 0.028), with AGA also having the highest levels.

A marginally significant difference was found for eosinophils (*p* = 0.052), with higher levels being detected in APA.

With reference to basophils, a marginally significant difference was also found (*p* = 0.051), having the control group the higher levels. About lymphocytes, no differences were found between groups (*p* = 0.253) (Table 2).

We evaluated via the Receiver Operating Characteristic (ROC) curve, the hematologic parameters, such as WBC, Neutrophil, and Monocytes (Figure 1), and the following parameters were determined as presented in the next paragraph.

(1)The area Under the Curve (AUC) for WBC: optimal sum of sensitivity (0.824) and specificity (0.308) at a cut-off level of greater than 10.150 × 10^9^/L is 0.767 (*p* = 0.001).(2)AUC for neutrophils: optimal sum of sensitivity (0.647) and specificity (0.154) at a cut-off level of greater than 9.360 × 10^9^/L is 0.753 (*p* = 0.001); (3) AUC for monocytes: optimal sum of sensitivity (0.765) and specificity (0.308) at a cut-off level of greater than 0.72 × 10^9^/L is 0.723 (*p* = 0.011).

### 2.3. Appendicular Lavage Fluid (ALF) Eosinophil Derived Neurotoxin (EDN), Eosinophil Cationic Protein (ECP), and Eosinophil Peroxidase (EP) Concentrations 

Appendicular Lavage Fluid EDN, ECP, and EP concentrations are presented in Table 3.

For EDN, no differences (*p* = 0.256) were found between groups (Table 3) and (Figure 2a).

Concerning ECP and EP significate differences were present (Table 3) (Figure 2b,c). For ECP, differences were found between groups (*p* < 0.001). Through multiple paired comparisons, the differences were found between the pair APA-Control (*p* = 0.004) and AGA-Control (*p* < 0.001), with the highest level of significance being recognized in the APA and AGA groups (Figure 2b). No differences were found between the groups APA and AGA (*p* = 0.176).

Regarding EP, differences were found between groups (*p* < 0.001); further, the differences were between APA-Control (*p* = 0.015) and AGA-Control (*p* < 0.001). Through pair-wise multiple comparisons, the highest levels of significance were related to the APA and AGA groups (Figure 2c). No differences were found between APA and AGA groups (*p* = 0.236).

In ALF determinations the lowest levels of eosinophil granule proteins were present in the control group.

### 2.4. Serum Eosinophil Derived Neurotoxin (EDN), Eosinophil Cationic Protein (ECP), and Eosinophil Peroxidase (EP) Concentrations

Serum EDN, ECP, and EP levels are depicted in Table 4.

Regarding EDN in peripheral blood, no significant differences were present (*p* = 0.067) (Table 4) (Figure 3).

Significant differences were found between histologic groups, regarding ECP (*p* < 0.001) and EP (*p* < 0.001) peripheral blood concentrations (Figure 4a,b).

Through pair-wise analysis, differences were verified for the pair APA-Control (*p* = 0.002) and the pair AGA-Control (*p* < 0.001), the highest levels of ECP associated with the APA and AGA groups. No difference was found between APA and AGA groups (*p* = 1.000).

Considering EP and by multiple comparisons analysis, the differences were found between APA-Control (*p* = 0.004) and AGA-Control (*p* < 0.001), with the highest levels being again associated with the APA and AGA groups. No difference was found between APA and AGA groups (*p* = 0.179).

In the serum, the lowest levels of eosinophil granule proteins were present in the control group.

We evaluated the ROC curve for ECP and EP (Figure 5).

### 2.5. Relationship between Eosinophil Derived Neurotoxin (EDN), Eosinophilic Cationic Protein (ECP), and Eosinophil Peroxidase (EP) Concentrations at Serum and Appendicular Lavage Fluid

The relationship between eosinophils granule proteins in the Serum and Appendicular Lavage Fluid is depicted in Table 5.

Significant correlations were found in the whole study analysis (*p* < 0.001) and between several pairs (i) ECP in PB with ECP in ALF determinations; (ii) EP in PB and ALF; (iii) ECP in ALF, with EP in PB and in ALF and (iv) EP in PB with EP in ALF (Table 5).

### 2.6. Relationship between Blood Eosinophils and Serum and Appendicular Lavage Fluid Eosinophil Derived Neurotoxin (EDN), Eosinophilic Cationic Protein (ECP), and Eosinophil Peroxidase (EP) Concentrations

The relationship between blood eosinophils and eosinophil granule proteins in the serum and Appendicular Lavage Fluid is depicted in Table 6.

### 2.7. Relationship between Serum Eosinophil Derived Neurotoxin (EDN), Eosinophilic Cationic Protein (ECP), Eosinophil Peroxidase (EP) Concentrations, and Acute Appendicitis Clinical Presentation

The relationship between serum EDN, ECP, EP levels, and appendicular perforation (Yes or No); peritonitis, presence (Yes) or absence (No), and appendicitis, complicated (Yes) or uncomplicated (No) is depicted in Table 7.

In any of the clinical settings evaluated, no differences were returned regarding EDN.

Concerning ECP, differences were found regarding appendicular perforation (*p* = 0.005) and the presence of peritonitis (*p* = 0.005). For complicated or uncomplicated appendicitis, no differences were found (*p* = 0.586).

Regarding EP, differences were found in the clinical setting of perforation and peritonitis, with the higher significance levels being related to perforation (*p* = 0.003) and the presence of peritonitis (*p* = 0.003). No differences were found for complicated/uncomplicated appendicitis.

For data with statistically significant differences, we evaluate by ROC curves the ECP and EP levels in the clinical setting of AA perforation (Figure 6).

ROC curves for ECP and EP levels and peritonitis are depicted in Figure 7. 

### 2.8. Relationship between Appendicular Lavage Fluid (ALF) Eosinophil Derived Neurotoxin (EDN), Eosinophilic Cationic Protein (ECP), Eosinophil Peroxidase (EP) Concentrations, and Acute Appendicitis Clinical Presentation

The relationship between Appendicular Lavage Fluid EDN, ECP, EP levels, and appendicular perforation (Yes or No); peritonitis, presence (Yes) or absence (No) and appendicitis, complicated (Yes) or uncomplicated (No) is depicted in Table 8.

The EDN values in ALF were not different in any of the clinical settings evaluated. Regarding ECP, differences were found in peritonitis (*p* = 0.002). Relatively to EP, the results were similar to ECP, the higher levels were found in patients with peritonitis (*p* = 0.021)—Table 8.

### 2.9. Other Data

There were 6 patients with perforation of the appendices (10%), 20 patients with peritonitis (33%), 19 with localized peritonitis, and 1 with generalized peritonitis.

Complicated AA, as defined previously, was present in 26 patients (46%).

## 3. Discussion

Epidemiological, clinical, histological, and experimental evidence suggests that an allergic component may occur in AA [16,26].

Conceptually AA can be a response to an external or internal antigen. The appendix has all the cell types that are linked to allergies. These cells are daily challenged with thousands of antigens [26]. The gut immune system must continuously distinguish innocuous dietary antigens and commensal microbes from pathogens [27].

Eosinophils participate in the allergic process by releasing proteins from their granules, such as ECP, EP, or EDN [28].

In the present study, we prospectively evaluated eosinophil granule protein concentrations at ALF and in blood, showing that ECP and EP levels are strongly elevated in AA, in comparison to the control group. A strong correlation was found between local and systemic levels for ECP and EP, with higher levels being traced back to ALF, which reflects a local intense inflammatory response. In fact, ECP is released locally, usually in the presence of an allergen, and subsequently, increases in circulation [29]. At appendicular slices, no helminth/parasites were observed, therefore, the elevation can be attributed to the presence of an unknown allergen.

We have reference values for several cationic eosinophil proteins serum concentrations. ECP serum levels in APA were 2.6 above the reference value, and in AGA they were 3.4 above the reference value. In APA, EP blood levels were 6.3 above the reference value and in AGA, results were even more pronounced, being 9.4 above the reference value. 

These results certainly suggest that an allergic component is involved in AA pathogenesis, resulting in an intense local reaction with repercussions at a systemic level.

The eosinophil proteins (ECP, EDN, and EP) have been evaluated in several biologic fluids, but never in ALF, so we do not have any reference values for comparison. 

For ECP, dose-dependent cytotoxic effects have been described, including necrosis apoptosis [30], and correlates with Chronic Obstructive Pulmonary Disease (COPD) and asthma severity [12,31]. Albeit, not statistically significant, serum and ALF ECP concentrations are higher in AGA than in APA, which can be reflective of disease severity, similar to their counterpart in COPD or asthma [12,31].

The literature is scarce in evaluating eosinophil granules protein and AA. A clinical study showed ECP serum elevation in AA, in comparison with healthy controls [32].

AA is a difficult clinical diagnosis, being imaging and laboratory data often used for diagnosis accuracy [33,34]. No inflammatory markers such as WBC counts or C-reactive proteins have been established as a singular marker ensuring high specificity or sensitivity in the diagnosis of appendicitis [24].

Our results corroborate the acceptable discriminatory capacity of WBC, neutrophils, and monocytes for the diagnosis of AA [35]. 

Concerning peripheral blood eosinophil count (PBEC), a marginally significant difference of higher count was found in APA. These results are in accordance with a Th2 response in AA, which is also observed in other allergic diseases [13,36].

In the present study no correlation was found between PBEC and local or systemic levels of any of the eosinophilic proteins studied. Santosh et al. also found no correlation between PBEC and serum ECP levels in patients with AA [37]. 

As blood levels can be determined and aid in the working-up diagnosis of acute appendicitis, we determine ROC curves for ECP and EP serum concentrations. Our results show that the discriminatory capacity of ECP serum concentrations to distinguish between the control group and the acute appendicitis group is good, and for EP serum concentrations, the discriminatory capacity is excellent. ECP and EP serum levels can be added to AA diagnostic scoring systems, as they are useful to clinically distinguish patients who had appendicitis from patients who do not have AA.

After AA diagnosis, which treatment should be proposed? Nonoperative management is considered a safe treatment for patients with uncomplicated AA. 

However, how to safely differentiate an uncomplicated AA from a complicated AA remains a challenge as there are no universally agreed guidelines for this [33].

We evaluated the relationship between eosinophilic protein serum concentrations and clinical consequences. Perforation is a situation with clinical relevance, being linked to higher morbidity, mortality rates, and length of stay [38,39]. 

We found that perforation was more common when ECP and EP systemic and ALF concentrations were higher, but their discriminatory power for the diagnosis of perforation is weak [35].

ECP and EP concentrations, both in blood and ALF, were higher in the presence of peritonitis, reflecting an intense inflammatory process. Their discriminatory power is acceptable for the diagnosis of peritonitis [35], but the clinical relevance of serum concentrations is minimal, as the diagnosis of peritonitis is clinical.

Complicated AA was defined by the presence of several clinical and pathologic events. Regarding the presence of complicated AA [34,40], no differences were found for any of the eosinophilic proteins, at both local and systemic concentrations.

When evaluated in separate clinical or pathologic settings, similar to perforation or peritonitis, the differences were statistically significant.

The etiology of AA remains poorly understood and bridging the knowledge gaps in the pathogenesis of appendicitis is necessary [2,5].

Our work shows that an allergic component is undoubtedly present in acute appendicitis and eosinophil granule protein concentration in the serum discriminates between AA and abdominal pain and helps identify perforation and peritonitis.

A novel area of research, involving allergy can be added to this common pathology, the AA, opening the way to innovative therapeutic arms in this pathology, traditionally treated by surgery.

Strengths: The prospective design of the study, with clear histologic confirmation of AA. This is the first time that eosinophil granule proteins were evaluated in the target organ of AA, having local inflammatory changes have been studied with an innovative concept, the appendicular lavage fluid, and then correlated with simultaneous determinations of systemic values.

Limitations: The sample size is small, and the procedure should be performed by others, ideally in a prospective, multicentre study, as this study was conceived as a single-center study. Eosinophil proteins are evaluated in biological fluids, never in NaCl 0.9%, such as in our case. Some of the proteins are highly unstable and despite measuring them as quickly as possible, we cannot guarantee that was the case for all the patients. The control group consisted of patients with colon cancer, a pathology that may influence the results and can be a possible bias factor. Furthermore, the age of the control group was significantly different from those of the AA group. It was not possible to obtain specimens from young adults, as a right colectomy for cancer patients is rare in this age group. Other alternatives will be in the context of accidental appendectomy in bladder and ovarian cancer, but the results could also be biased, for the presence of oncological pathology, and they will be also older. Obtaining appendices in other circumstances will be hard and unethical. Patients who operated during the night were excluded and this may have biased the results.

## 4. Materials and Methods

### 4.1. Aim

The primary aim of this study is to evaluate the participation of eosinophil granule proteins in acute appendicitis, both at local and systemic levels and so, evaluate for the presence of a hypersensitivity type 1 allergic reaction.

The secondary aim is to evaluate the diagnostic accuracy of eosinophil granule proteins for the detection of acute appendicitis, as well as for distinguishing between complicated and uncomplicated acute appendicitis.

### 4.2. Patients and Study Design

All patients over the age of 18 admitted between August 2021 to April 2022, with a clinical picture compatible with AA and submitted to appendectomy, were eligible for this study.

Pregnant women were excluded, as well as under-18-year-old patients, because these patients are being taken care of by the pediatrics department.

Patients with either a history of hyper-eosinophilic disease, such as hyper-eosinophilic syndrome, eosinophilic granulomatosis, eosinophilic esophagitis/gastritis, or receiving oral corticosteroids, anti-eosinophilic therapies within, respectively, the past 1 and 6 months, were also excluded [41].

Patients operated between 9 PM and 8 AM were also excluded because in that schedule there were no laboratory facilities for eosinophilic granule protein determinations.

At an interim analysis, all the patients submitted to appendectomy have histologic features of AA, then the possibility of recruiting patients for the control group was accepted to be hampered. After approval by the Ethics Committee (Adenda 82/2021), the control group was extended to patients submitted to the right colectomy, and appendectomy was performed in the right colectomy specimens [42].

### 4.3. Setting

This prospective single-center study was performed in a 600-bed tertiary public hospital, which provides medical care to a 280,000 urban habitats population. Both the Surgical and Pathology departments are Associate Academic Centers (Faculdade de Medicina da Universidade de Lisboa). Joaquim Chaves Lab is a private laboratory that has a working protocol with Garcia de Orta Hospital.

### 4.4. Appendicular Lavage Fluid

After appendicectomy, a gauge was inserted in the appendix proximal luminal aspect and 3 mL of saline 0.9% was instilled and collected for ALF. The process was performed three times in a standardized fashion [25]. In the control group, appendectomy was performed in right colectomy specimens and the ALF procedure accomplish.

Training on the steps of the procedure was provided, to all intervenient, before starting the study.

### 4.5. Pathologic Analysis

After macroscopic evaluation, the appendix was cut into three slices, at the base, tip, and middle portion. The slices were embedded in paraffin and colorized with hematoxylin and eosin afterwards.

Appendicitis was classified according to appendicular wall neutrophil distribution: acute phlegmonous or suppurative appendicitis (APA) was defined as the presence of neutrophilic infiltration in *muscular propria* and acute gangrenous appendicitis (AGA) as necrosis of the appendicular wall [43,44]. The presence of neutrophils at the mucosa level was considered a normal distribution with no clinical significance [43]. The specimens were classified as normal histologic findings (NHF) when no neutrophil infiltrate was shown in *muscular propria* [43].

All the appendicular specimens were evaluated by a general pathologist and then revised by a dedicated gastrointestinal pathologist (CH), blinded to the previous report. 

On the rare occasions when there was no agreement in the specimen evaluation, a consensus was reached.

### 4.6. Laboratory Proceedings

#### 4.6.1. Hemogram

Blood samples were obtained from all patients after admission to the emergency ward. Routinely performed white blood cell (WBC) counts included the following mature leukocyte subpopulations: eosinophils, neutrophils, lymphocytes, basophils, and monocytes [32]. Measures were made by an automated hematology analyzer DxH 900 equipment (Beckman Coulter, Inc, Brea, CA, USA.), using the Coulter principle and standards. For evaluating the WBC population with a differential test, VCSn technology was used, being the results automatically expressed in mm^3^ in peripheral blood. The count is made in triplicate to assure the security and reproducibility of the results.

Reference intervals for WBC count are 4.0–11.0 × 10^9^/L, for neutrophil, reference interval and % of WBC are 1.90–8.00 × 10^9^/L (40–70%), for lymphocyte are 0.9–5.20 × 10^9^/L (19–48%), for monocytes are 0.16–1.00 × 10^9^/L (3.4–9.0%), for eosinophil are 0.00–0.80 × 10^9^/L (0.0–7.0%), and for basophil counts are 0.00–0.20 × 10^9^/L (0.0–1.5%).

#### 4.6.2. Eosinophil-Derived Neurotoxin (EDN), Eosinophilic Cationic Protein (ECP), and Eosinophil Peroxidase (EP) Determinations

Blood was collected in vacutainers before anesthesia induction by venepuncture. Plasma was obtained from EDTA-anticoagulated blood, kept at 2–8 °C, and centrifugated an hour after the phlebotomy. ALF was kept at 2–8 °C and centrifuged at 3000 rpm as quickly as possible after collection. Only the supernatant was saved and used for analysis. Serum, plasma, and appendicular washes were aliquoted, stored at −70 °C, and thawed once just before being assayed for mediator levels. For EDN and EP, serum and ALF were assayed by a commercial ELISA kit, according to the manufacturer’s instructions (MyBioSource Inc., San Diego, CA, USA). For ECP levels, serum and ALF were assayed by a commercial ELISA kit, according to the manufacturer’s instructions (DRG Instruments GmbH, Marburg Germany).

Serum normal values for ECP are <15.6 ng/mL, for EDN < 5 ng/mL, and for EP < 25 ng/mL. Normal values for ALF are unknown, as this is the first report on the subject.

### 4.7. Other Definitions

Appendicular perforation was defined by gangrenous alterations and a transmural defect in the appendicular wall, or by the presence of a fecalith in the abdominal cavity [37,45].

Peritonitis was defined by the operating surgeon as an inflammatory exudation on the peritoneum; in addition to being defined as localized, it is further when one or two quadrants were involved, or as generalized, when more than two quadrants were involved [46].

Despite the clinical importance of distinguishing complicated from uncomplicated appendicitis, no universally agreed definition exists on how to classify each one [33]. 

Complicated appendicitis was defined as the presence of perforation, gangrenous appendicitis (histological criteria), peritonitis, appendicular plastron, and pelvic or intra-abdominal abscess [34,40]. Uncomplicated appendicitis was defined as an inflamed appendix, in the absence of gangrene, perforation, or abscess around the appendix [34].

### 4.8. Ethics

The study was approved by the Ethics Committee of the Hospital Garcia de Orta (Centro Garcia de Orta, Reference Number 82/2021; date of approval: 23 June 2021). An interim analysis showed that all the appendectomy specimens had histologic criteria of AA, which means the absence of a control group. So, an addendum was solicited to include patients submitted to right colectomy and the ALF was performed at the appendectomy specimen of colectomy (Centro Garcia de Orta, Adenda 82/2021; date of approval: 11 November 2021). 

Patients received oral information before leaving their written consent of participation in the study. All data was pseudo-anonymized and the results were presented in such a way that made it impossible to identify single patients

### 4.9. Statistical Analysis

Our study is the first of this kind, there was no prior data or expectations on differences (either if they exist or their magnitude) and so, it was then difficult, or even impossible, to determine a good sample size. We, therefore, collected all the data that we could in a period of 10 months and did the analysis that we now present.

Data were analyzed using SPSS statistical software, V27.0 for Windows. The results were considered significant at the 5% significant level. To test the normality of the data, the Shapiro-Wilk test was used. To characterize the sample, frequency analysis (n, %) was used for qualitative data. Median [Percentile 25%–Percentile 75% (Q1–Q3)] was used for quantitative data. To compare age and BMI, WBC, neutrophils, lymphocytes, basophils, eosinophils, monocytes, EDN, ECP, and EP between the appendicular specimen’s histology (in the three study groups, Control, APA, and AGA), the One-Way ANOVA (normality assumption verified) or the Kruskal-Wallis test (normality assumption not verified) were used. When statistically significant differences were detected, Tuckey-HSD (normality assumption verified) or Kruskal-Wallis multiple comparison tests (normality assumption not verified) were used for pair-wise analysis.

To compare the levels of EDN, ECP, EP, and the occurrence of perforation, peritonitis, and complicated appendicitis, a *t*-test (normality assumption verified) or Mann-Whitney test (normality assumption not verified) was used.

To verify whether the distribution of gender was homogeneous among the three studied groups, the Chi-Square test was used, and for allergies (presence/absence) the Chi-Square test by Monte Carlo simulation was used since the assumptions of applicability of the Chi-Square test was not verified.

ROC analysis was performed to evaluate WBC, monocytes, and neutrophils performances for the diagnosis of AA and the performance of ECP and EP for the diagnosis of simple and complicated appendicitis. The AUC varies between 0 and 1. The present indicative values used to classify the discriminant power are: 0.5—no discriminative; [0.5, 0.7] weak; [0.7, 0.8] acceptable; [0.8, 0.9] good and ≥0.9 excellent [35].

## 5. Conclusions

Eosinophilic cationic protein and eosinophil peroxidase concentrations are elevated in serum and appendicular lavage fluid in patients with acute appendicitis. This data corroborates the presence of an allergic component in acute appendicitis, as these cationic eosinophilic proteins are involved in a Th2 immune response. 

ECP and EP serum concentrations can be added to AA diagnostic scoring systems, as they may be helpful to distinguish patients who had appendicitis from patients with non-specific acute abdominal pain.

## Figures and Tables

**Figure 1 ijms-24-09091-f001:**
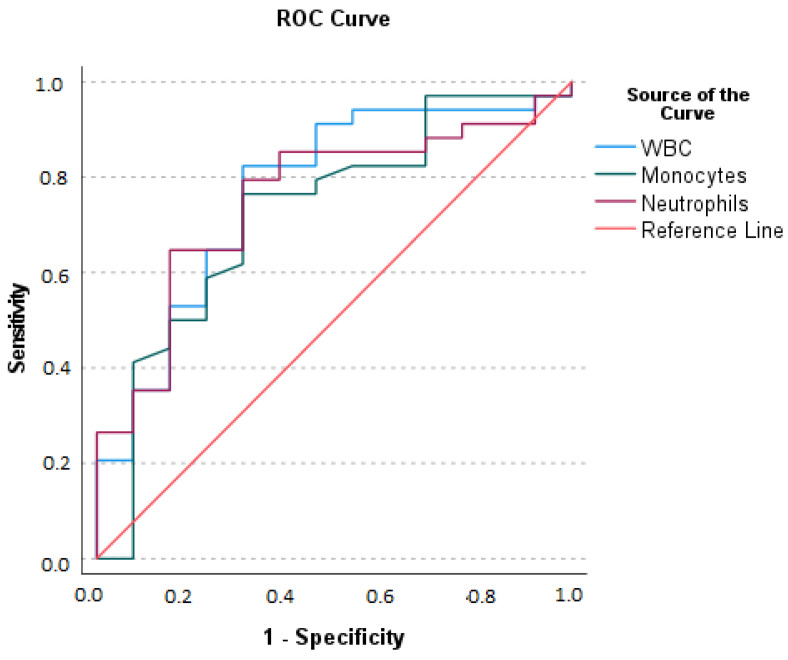
Receiving Operating Characteristics (ROC) curves for White Blood Cells (WBC), neutrophils, and monocytes for the diagnosis of Acute Appendicitis.

**Figure 2 ijms-24-09091-f002:**
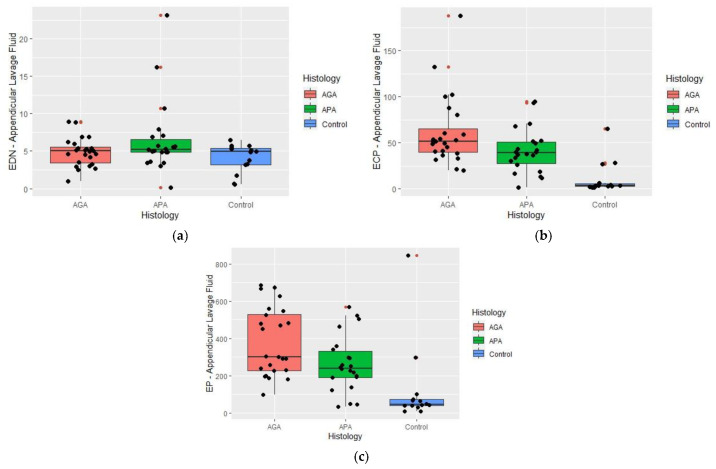
(**a**–**c**) Box-plot summarizing Eosinophil Derived Neurotoxin (EDN), Eosinophil Cationic Protein (ECP), and Eosinophil Peroxidase (EP) concentrations (ng/mL) in Appendicular Lavage Fluid (ALF) from histologically proven normal appendices (Control), acute phlegmonous (APA) and gangrenous appendicitis (AGA) groups. Median values and interquartile ranges are denoted by horizontal bars and boxes. Outliers are represented by the red spot • and observed values by a black spot •. (**a**) EDN appendicular lavage fluid concentrations and histology; (**b**) ECP appendicular lavage fluid and histology; (**c**) EP appendicular lavage fluid concentrations and histology.

**Figure 3 ijms-24-09091-f003:**
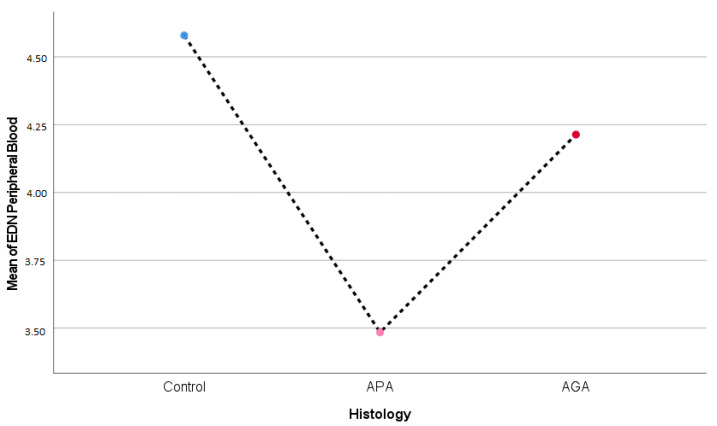
EDN (Eosinophil Derived Neurotoxin) blood peripheral concentrations (ng/mL) and appendicular histology.

**Figure 4 ijms-24-09091-f004:**
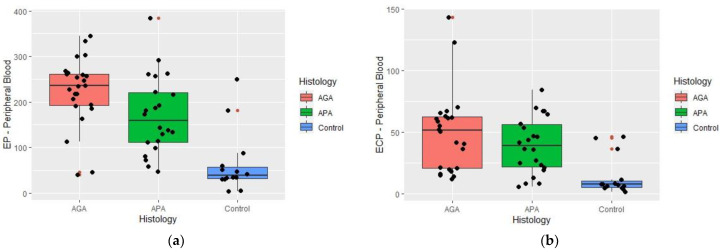
Box-plot summarizing Eosinophil Cationic Protein (ECP) and Eosinophil Peroxidase (EP) blood concentrations (ng/mL) in a histologically proven normal appendix (control), acute phlegmonous (APA), and gangrenous appendicitis (AGA) groups. Median values and interquartile ranges are denoted by horizontal bars and boxes. Outliers are represented by a red spot • and observed values by a black spot •. (**a**) ECP blood concentrations and appendicular histology were appendicular; (**b**) EP blood concentrations and appendicular histology.

**Figure 5 ijms-24-09091-f005:**
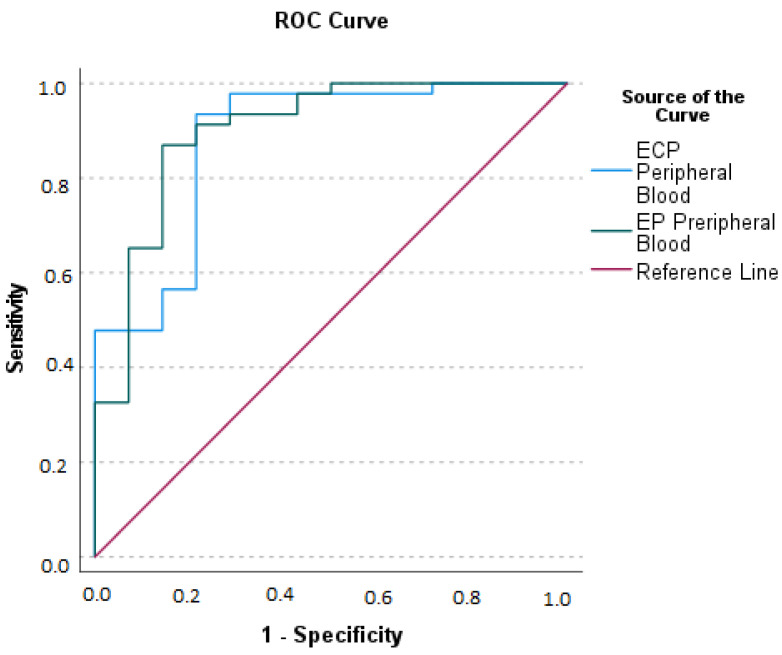
Receiving Operating Characteristics (ROC) curves for ECP (Eosinophil Cationic Protein) and EP (Eosinophil Peroxidase) peripheral blood concentrations of patients with Acute Appendicitis. Area Under the Curve (AUC) for ECP: optimal sum of sensitivity (0.935) and specificity (0.214) at a cut-off level of greater than 11.41 ng/mL is 0.880 (*p* < 0.001). AUC for EP: optimal sum of sensitivity (0.870) and specificity (0.143) at a cut-off level greater than 93.20 ng/mL is 0.901 (*p* < 0.001).

**Figure 6 ijms-24-09091-f006:**
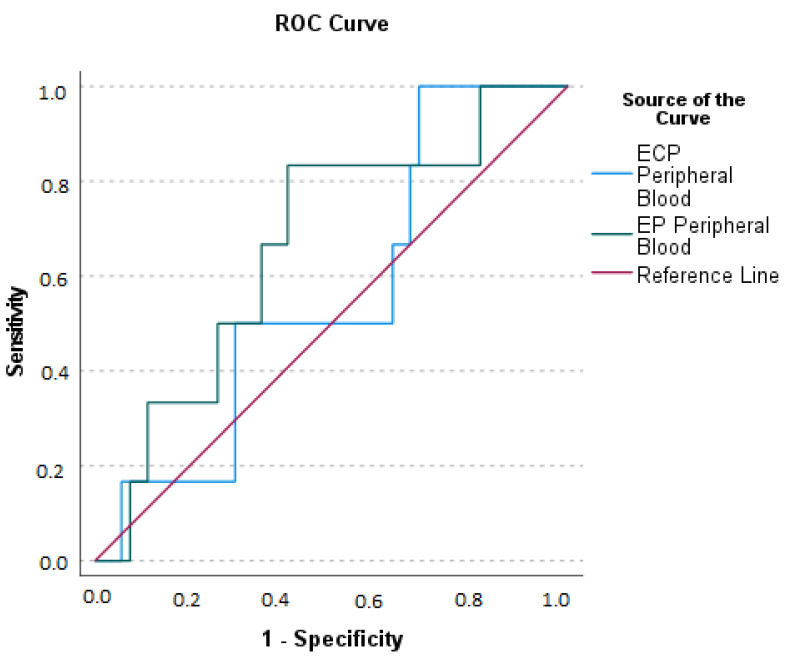
Receiving Operating Characteristics (ROC) curves for ECP (Eosinophil Cationic Protein) and EP (Eosinophil Peroxidase) for the diagnosis of Acute Appendicitis with perforation. Area Under the Curve (AUC) for ECP: the optimal sum of sensitivity (1.00) and specificity (0.685) at a cut-off level of greater than 13.37 ng/mL is 0.562 (*p* = 0.595). AUC for EP: the optimal sum of sensitivity (0.833) and specificity (0.407) at a cut-off level greater than 188.99 ng/mL is 0.664 (*p* = 0.163).

**Figure 7 ijms-24-09091-f007:**
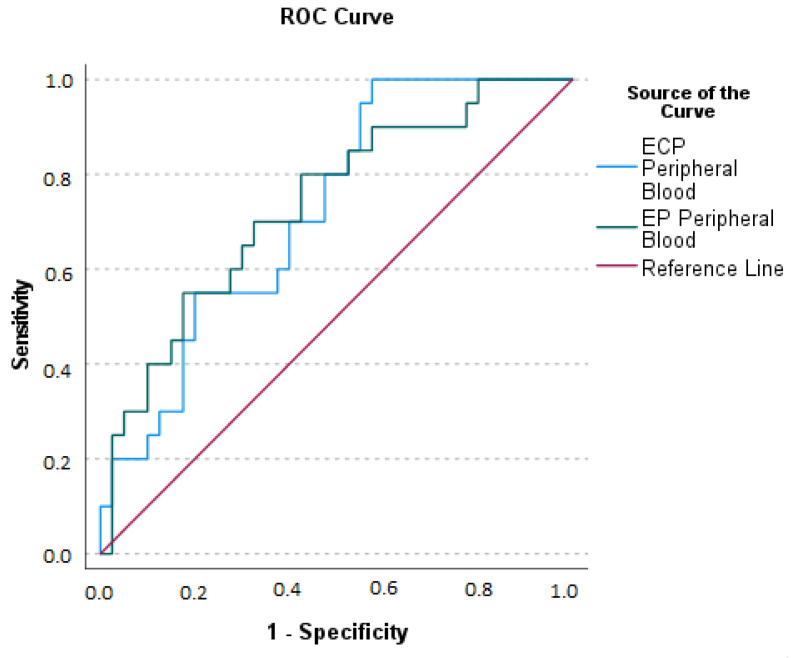
Receiving Operating Characteristics (ROC) curves for ECP (Eosinophil Cationic Protein) and EP (Eosinophil Peroxidase) for the diagnosis of Acute Appendicitis with peritonitis. Area Under the Curve (AUC) for ECP: the optimal sum of sensitivity (1.00) and specificity (0.575) at a cut-off level of greater than 13.37 ng/mL is 0.724 (*p* = 0.001). AUC for EP: the optimal sum of sensitivity (0.550) and specificity (0.175) at a cut-off level greater than 235.27 ng/mL is 0.735 (*p* = 0.001).

**Table 1 ijms-24-09091-t001:** Study groups and patient demographics.

	Control	APA	AGA	*p*-Value
N (%)	14 (23.3)	22 (36.7)	24 (40)	
Age (y)	67.5 (30–78)	48 (36–66)	31 (23–55)	0.044 *
Sex M/F	9/5 (64.3/35.7)	11/11 (50/50)	12/12 (50/50)	0.644 **
Allergy N/Y	12/1 (92.3/7.7)	18/2 (90/10)	20/1 (95.2/4.8)	0.834 ***
BMI	28.04 ± 4.76	24.57 ± 4.07	26.44 ± 4.540	0.298 ****

APA—Acute Phlegmonous Appendicitis; AGA—Acute Gangrenous Appendicitis; M—Male; F—Female; N—No; Y—Yes; BMI-Body Mass Index, Kg/m^2^; Results are presented Mean ± SD and Median (Q1–Q3). Sex in number and (%); * Kruskal-Wallis test. ** Qui-Square test. *** Qui-Square test by Monte Carlo Simulation **** One-way ANOVA.

**Table 2 ijms-24-09091-t002:** Hemogram and appendicular histology.

	Control	APA	AGA	*p*-Value
WBC	10.10 ± 3.43	12.08 ± 2.93	15.97 ± 5.05	<0.001 **
Neutrophils	6.73 (5.69–9.28)	8.21 (7.20–12.46)	11.50 (9.50–14.70)	0.005 *
Lymphocytes	1.29 (1.05–1.86)	1.83 (1.63–2.42)	1.37 (0.83–2.10)	0.253 *
Basophils	0.06 ± 0.02	0.03 ± 0.02	0.04 ± 0.02	0.051 **
Eosinophils	0.04 (0.01–0.21)	0.14 (0.04–0.26)	0.02 (0.00–0.07)	0.052 *
Monocytes	0.06 (0.04–0.07)	0.03 (0.02–0.05)	0.03 (0.02–0.06)	0.034 *

APA—Acute Phlegmonous Appendicitis; AGA—Acute Gangrenous Appendicitis. WBC—White Blood Count; WBC and basophils are expressed in absolute numbers × 10^9^/L, mean ± SD; Neutrophils, lymphocytes, eosinophils, and monocytes are expressed in absolute numbers × 10^9^/L, Median(Q1–Q3) * Kruskal-Wallis test. ** One Way-ANOVA. Statistically significant differences at a 5% significance level.

**Table 3 ijms-24-09091-t003:** Eosinophils Granule Proteins Levels (ALF) and Appendicular Histology.

	Control	APA	AGA	*p*-Value
EDN	4.93 (3.20–5.43)	5.23 (4.88–6.9)	5.02 (3.55–5.65)	0.256 *
ECP	3.35 (2.60–6.10)	38.85 (26.50–51.77)	51.55 (39.55–70.09)	<0.001 *
EP	45.55 (39.9–74.0)	240.28 (191.2–341.3)	302.5 (227.7–535.85)	<0.001 *

ALF—Appendicular Lavage Fluid. APA—Acute Phlegmonous Appendicitis; AGA—Acute Gangrenous Appendicitis. EDN—Eosinophil-derived neurotoxin, ECP—eosinophil cationic protein (ECP), EP-Eosinophil peroxidase, Measurements are presented as ng/mL. Results are presented Median (Q1–Q3) * Kruskal-Wallis test. Statistically Significant differences at a 5% significance level.

**Table 4 ijms-24-09091-t004:** Eosinophils Granule Proteins Levels (PB) and Appendicular Histology.

	Control	APA	AGA	*p*-Value
EDN	4.58 ± 1.63	3.48 ± 1.13	4.21 ± 1.54	0.067 **
ECP	7.56 (4.76–11.13)	39 (21.30–56.90)	51.30 (20.25–62.59)	<0.001 *
EP	38.20 (30.20–59.10)	158.4 (111.09–222.1)	235.27 (192.33–262.51)	<0.001 *

PB—peripheral blood; APA—Acute Phlegmonous Appendicitis; AGA—Acute Gangrenous Appendicitis; EDN—Eosinophil-derived neurotoxin, ECP—eosinophil cationic protein (ECP); EP—Eosinophil peroxidase; Measurements are presented as ng/mL; Results are presented as Mean ± SD or Median(Q1–Q3); * Kruskal-Wallis test. ** One-way ANOVA Statistically Significant differences at a 5% significance level.

**Table 5 ijms-24-09091-t005:** Eosinophils Granule Proteins: Blood and ALF Correlations.

	ECP (ALF)	EDN (PB)	EDN (ALF)	EP (PB)	EP (ALF)
r_S_ (*p*-Value)					
ECP (PB)	0.767 ** (<0.001)	0.137 (0.288)	0.223 (0.082)	0.676 ** (<0.001)	0.544 ** (<0.001)
ECP (ALF)		0.142 (0.271)	0.128 (0.322)	0.726 ** (<0.001)	0.671 ** (<0.001)
EDN (PB)			0.087 (0.500)	0.095 (0.462)	0.158 (0.221)
EDN (ALF)				0.150 (0.243)	0.072 (0.581)
EP (PB)					0.643 ** (<0.001)

PB—Peripheral Blood; ALF—Appendicular Lavage Fluid; EDN—Eosinophil-derived neurotoxin, ECP—eosinophil cationic protein; EP-Eosinophil peroxidase; r_S_: Spearman correlation coefficient; ** Correlation is significant at the 0.01 level (2-tailed).

**Table 6 ijms-24-09091-t006:** Correlation between Blood Eosinophils and PB and ALF Eosinophils Granules proteins: EDN, ECP, and EP.

			Eosinophils
EDN	PB	r_S_ (*p*-value)	−0.274 (0.054)
	ALF		−0.034 (0.812)
ECP	PB	r_S_ (*p*-value)	−0.148 (0.304)
	ALF		−0154 (0.287)
EP	PB	r_S_ (*p*-value)	−0.0146 (0.311)
	ALF		−0.217 (0.129)

PB—Peripheral blood, ALF—Appendicular Lavage Fluid; EDN-Eosinophil-derived neurotoxin, ECP—eosinophil cationic protein; EP—Eosinophil peroxidase; r_S_—Spearman Correlation Coefficient (*p*-value) No correlation was found between these determinations.

**Table 7 ijms-24-09091-t007:** Peripheral Blood Eosinophilic Granules Proteins and Clinic.

		EDN	*p*-Value	ECP	*p*-Value	EP	*p*-Value
Perforation	No	4.35 (2.9–5.04)	0.667	35.95 (8.4–58.7)	0.005	168 (58.19–250.47)	0.003
	Yes	4.73 (3.9–5.09)		35 (16.20–52.10)		232.15 (191–264.2)	
Peritonitis	No	4.15 (2.9–5.03)	0.527	21.05 (7.84–46.2)	0.005	132.02 (47.5–220.5)	0.003
	Yes	4.6 (3.05–5.14)		51.30 (22.2–64.3)		241.6 (168–278.35)	
Complicated	No	3.14 (2.8–4.53)	0.119	42.77 (16–59.07)	0.586	164.93 ± 90.13	0.08 *
	Yes	4.6 (2.91–5.09)		41.02 (21–61.20)		213.55 ± 81.18	

EDN—Eosinophil-derived neurotoxin, ECP—eosinophil cationic protein; EP—Eosinophil peroxidase; Measurements are presented in ng/mL; Results are presented Median (Q1–Q3); Mann-Whitney was used to analyze differences between groups, except * *T*-test.

**Table 8 ijms-24-09091-t008:** Appendicular Lavage Fluid Eosinophilic Granules Proteins and Clinic.

		EDN	*p*-Value *	ECP	*p*-Value	* EP	*p*-Value *
Perforation	No	4.63 (2.43–5.6)	0.538	4.92 (11.8–52.6)	0.475	274.8 (68.5–463.4)	0.961
	Yes	5.14 (4.5–5.92)		48.5 (31.9–54.5)		276.4 (188.3–257.9)	
Peritonitis	No	4.81 (3.3–5.5)	0.259	31.06 (4.05–50.3)	0.002	235.98 (50.1–322.4)	0.021
	Yes	4.6 (3.05–5.14)		51.30 (22.2–64.3)		241.6 (168–278.35)	
Complicated	No	4.98 (3.5–5.5)	0.854	38.75 (22.34–50.4)	0.07	223.5 (159.5–330.3)	0.059
	Yes	5.08 (3.48–5.92)		50.5 (36.5–60.09)		292.4 (226–584.76)	

EDN—Eosinophil-derived neurotoxin, ECP—eosinophil cationic protein; EP—Eosinophil peroxidase; Measurements are presented in ng/mL; Results are presented Median (Q1–Q3) * Mann-Whitney was used to analyze differences between groups.

## Data Availability

Not applicable.

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
