# Peer review of "Eosinophil Granule Proteins Involvement in Acute Appendicitis—An Allergic Disease?"

_ijms, 2023, doi:10.3390/ijms24109091_

Round 1
Reviewer 1 Report
The authors evaluated the putative role of eosinophil granule proteins degranulated in acute appendicitis, both at a systemic level, in peripheral blood, and at the local level, in the appendix. They concluded that eosinophilic cationic protein and eosinophil peroxidase are elevated in serum and appendicular lavage fluid in acute appendicitis. This data corroborates the presence of an allergic component in acute appendicitis, as these cationic eosinophilic proteins are involved in a Th2 immune response.
I read the study with interest. Although the study is interesting and important, several major points should be addressed/corrected before any favorable decision is made. My concerns are as follows:
1. Abstract – Please provide exact results together with p-values, not only statements that something is increased/decreased.
2. Introduction is poor and not informative. The authors should add a few lines regarding different biomarkers that were recently investigated. I would suggest adding the following text and references:
In addition to standard biomarkers several new biomarkers for acute appendicitis such as hyponatremia, hyperfibrinogenemia, hyperbilirubinemia, leucine-rich α-2-glycoprotein 1, neutrophil gelatinase-associated lipocalin (NGAL), or interleukin-6 (IL-6) have been recently investigated and showed good predictive values for the detection of acute appendicitis, as well as for distinguishing between complicated and simple acute appendicitis. In addition, a more recent study showed that non-invasive markers from saliva, such as leucine-rich α-2-glycoprotein 1 (LRG1) may be a promising biomarker for the detection of acute appendicitis in children.
· Arredondo Montero J, Antona G, Bardají Pascual C, Bronte Anaut M, Ros Briones R, Fernández-Celis A, Rivero Marcotegui A, López-Andrés N, Martín-Calvo N. Serum neutrophil gelatinase-associated lipocalin (NGAL) as a diagnostic tool in pediatric acute appendicitis: a prospective validation study. Pediatr Surg Int. 2022;38(11):1569-1576. doi 10.1007/s00383-022-05197-w.
· Arredondo Montero J, Bardají Pascual C, Bronte Anaut M, López-Andrés N, Antona G, Martín-Calvo N. Diagnostic performance of serum interleukin-6 in pediatric acute appendicitis: a systematic review. World J Pediatr. 2022;18(2):91-99. doi: 10.1007/s12519-021-00488-z.
· Tintor G, Jukić M, Šupe-Domić D, Jerončić A, Pogorelić Z. Diagnostic Utility of Serum Leucine-Rich α-2-Glycoprotein 1 for Acute Appendicitis in Children. Journal of Clinical Medicine. 2023; 12(7):2455
· Wu Z, Zhao L, Feng S, Luo J. Hyperfibrinogenemia and hyponatremia as predictors of perforated appendicitis in children: A retrospective cohort study. Int J Colorectal Dis. 2023;38(1):72. doi: 10.1007/s00384-023-04362-4
· Kakar M, Delorme M, Broks R, Asare L, Butnere M, Reinis A, Engelis A, Kroica J, Saxena A, Petersons A. Determining acute complicated and uncomplicated appendicitis using serum and urine biomarkers: interleukin-6 and neutrophil gelatinase-associated lipocalin. Pediatr Surg Int. 2020;36(5):629-636. doi 10.1007/s00383-020-04650-y.
· Pogorelić Z, Lukšić AM, Mihanović J, Đikić D, Balta V. Hyperbilirubinemia as an Indicator of Perforated Acute Appendicitis in Pediatric Population: A Prospective Study. Surg Infect (Larchmt). 2021 Dec;22(10):1064-1071. doi: 10.1089/sur.2021.107.
· Pogorelić Z, Lukšić B, Ninčević S, Lukšić B, Polašek O. Hyponatremia as a predictor of perforated acute appendicitis in the pediatric population: A prospective study. J Pediatr Surg. 2021;56(10):1816-1821. doi: 10.1016/j.jpedsurg.2020.09.066.
· Tintor G. et al. Diagnostic accuracy of leucine-rich α-2-glycoprotein 1 as a non-invasive salivary biomarker in pediatric appendicitis. Int J Mol Sci. 2023;24:6043. doi: 10.3390/ijms24076043
3. Paragraph 2.1. The authors stated that the main demographic characteristics of the enrolled group of 60 patients and the 14 control patients were reflected in Table 1. It should be 46 patients and 14 patients from the control group. Please revise.
4. The resolution of all tables is poor. The authors should insert tables from MS Word, not cropped, because the resolution is not good. Please revise accordingly!
5. The discussion section is somehow messy and needs to be revised/restructured. Do not provide a review of the literature in this section. Do not discuss your results piecemeal. Focus on the results of the main objectives of the study. Write in four consecutive paragraphs (without headings): (i) summary (not data) of the findings of this study; (ii) logical and coherent comparison with the existing literature, focusing on the main objective(s); (iii) limitations of the study; and (iv) implications for practice/policy/research with a concluding statement.
6. Control group – The choice of a control group may be one of the major problems of this study. The authors choose patients with colon cancer. Such a type of pathology may significantly influence the results. The authors should emphasize under the limitations of the study that the presence of cancer in the control group may be one of the possible factors for bias.
7. The authors should include the date of approval together with the Institutional Review Board (IRB) approval reference.
8. At least two pathologists should interpret the results per objective criteria in such studies. As I can see only one pathologist reviewed all specimens.
9. The primary and secondary outcomes of the study should be clearly stated in the methodology.
10. The sample size presented in this study is small. Because this is a prospective study, sample size calculation is mandatory. It is unclear how the authors calculated the sample size. How did they decide how many patients will be included in each study group? The methodology does not include a sample size calculation, which is one of the most important parameters for the study design. How do the authors know that the present sample size is sufficient to achieve statistical significance? This should be described in detail.
11. The quality of the English language is below the standard and should be significantly improved. The manuscript would benefit from editing by a native English speaker or a professional language editor to improve grammar and readability.
The quality of the English language is below the standard and should be significantly improved. The manuscript would benefit from editing by a native English speaker or a professional language editor to improve grammar and readability
Author Response
We would like to thank the reviewers for their detailed comments and constructive suggestions to improve our manuscript. We have addressed all of the reviewer’s comments and concerns, hoping that our manuscript is now suitable for publication in this journal.
- Abstract – Please provide exact results together with p-values, not only statements that something is increased/decreased.
Response 1. The abstract has been reformulated. The exact results with p and AUC values were added in the abstract
- Introduction is poor and not informative. The authors should add a few lines regarding different biomarkers that were recently investigated. I would suggest adding the following text and references:
In addition to standard biomarkers several new biomarkers for acute appendicitis such as hyponatremia, hyperfibrinogenemia, hyperbilirubinemia, leucine-rich α-2-glycoprotein 1, neutrophil gelatinase-associated lipocalin (NGAL), or interleukin-6 (IL-6) have been recently investigated and showed good predictive values for the detection of acute appendicitis, as well as for distinguishing between complicated and simple acute appendicitis. In addition, a more recent study showed that non-invasive markers from saliva, such as leucine-rich α-2-glycoprotein 1 (LRG1) may be a promising biomarker for the detection of acute appendicitis in children.
- Arredondo Montero J, Antona G, Bardají Pascual C, Bronte Anaut M, Ros Briones R, Fernández-Celis A, Rivero Marcotegui A, López-Andrés N, Martín-Calvo N. Serum neutrophil gelatinase-associated lipocalin (NGAL) as a diagnostic tool in pediatric acute appendicitis: a prospective validation study. Pediatr Surg Int. 2022;38(11):1569-1576. doi 10.1007/s00383-022-05197-w.
- Arredondo Montero J, Bardají Pascual C, Bronte Anaut M, López-Andrés N, Antona G, Martín-Calvo N. Diagnostic performance of serum interleukin-6 in pediatric acute appendicitis: a systematic review. World J Pediatr. 2022;18(2):91-99. doi: 10.1007/s12519-021-00488-z.
- Tintor G, Jukić M, Šupe-Domić D, Jerončić A, Pogorelić Z. Diagnostic Utility of Serum Leucine-Rich α-2-Glycoprotein 1 for Acute Appendicitis in Children. Journal of Clinical Medicine. 2023; 12(7):2455
- Wu Z, Zhao L, Feng S, Luo J. Hyperfibrinogenemia and hyponatremia as predictors of perforated appendicitis in children: A retrospective cohort study. Int J Colorectal Dis. 2023;38(1):72. doi: 10.1007/s00384-023-04362-4
- Kakar M, Delorme M, Broks R, Asare L, Butnere M, Reinis A, Engelis A, Kroica J, Saxena A, Petersons A. Determining acute complicated and uncomplicated appendicitis using serum and urine biomarkers: interleukin-6 and neutrophil gelatinase-associated lipocalin. Pediatr Surg Int. 2020;36(5):629-636. doi 10.1007/s00383-020-04650-y.
- Pogorelić Z, Lukšić AM, Mihanović J, Đikić D, Balta V. Hyperbilirubinemia as an Indicator of Perforated Acute Appendicitis in Pediatric Population: A Prospective Study. Surg Infect (Larchmt). 2021 Dec;22(10):1064-1071. doi: 10.1089/sur.2021.107.
- Pogorelić Z, Lukšić B, Ninčević S, Lukšić B, Polašek O. Hyponatremia as a predictor of perforated acute appendicitis in the pediatric population: A prospective study. J Pediatr Surg. 2021;56(10):1816-1821. doi: 10.1016/j.jpedsurg.2020.09.066.
- Tintor G. et al. Diagnostic accuracy of leucine-rich α-2-glycoprotein 1 as a non-invasive salivary biomarker in pediatric appendicitis. Int J Mol Sci. 2023;24:6043. doi: 10.3390/ijms24076043
Response 2. The text and references were added, and the introduction was reformulated with more informative details.
- Paragraph 2.1. The authors stated that the main demographic characteristics of the enrolled group of 60 patients and the 14 control patients were reflected in Table 1. It should be 46 patients and 14 patients from the control group. Please revise.
Response 3. The paragraph was revised and the correct numbers of patients and control were applied.
- The resolution of all tables is poor. The authors should insert tables from MS Word, not cropped, because the resolution is not good. Please revise accordingly!
Response 4. Tables were inserted in MS Word
- The discussion section is somehow messy and needs to be revised/restructured. Do not provide a review of the literature in this section. Do not discuss your results piecemeal. Focus on the results of the main objectives of the study. Write in four consecutive paragraphs (without headings): (i) summary (not data) of the findings of this study; (ii) logical and coherent comparison with the existing literature, focusing on the main objective(s); (iii) limitations of the study; and (iv) implications for practice/policy/research with a concluding statement.
Response 5. The discussion has been extensively reviewed according to revisers proposal.
- Control group – The choice of a control group may be one of the major problems of this study. The authors choose patients with colon cancer. Such a type of pathology may significantly influence the results. The authors should emphasize under the limitations of the study that the presence of cancer in the control group may be one of the possible factors for bias.
Response 6. The possible bias factor and discussion was introduced in limitations
- The authors should include the date of approval together with the Institutional Review Board (IRB) approval reference.
Response 7. The study was approved by the Ethics Committee of the Hospital Garcia de Orta (Centro Garcia de Orta, Reference Number 82/2021, 23/6/2021) and Centro Garcia de Orta, Adenda 82/2021; 11/11/2021
- At least two pathologists should interpret the results per objective criteria in such studies. As I can see only one pathologist reviewed all specimens.
Response 8. The specimen was initially evaluated by a general pathologist. and then revised by a dedicated gastrointestinal pathologist (CH), blinded to the previous report. On the rare occasions when there was no agreement in the specimen evaluation, a consensus was reached.
- The primary and secondary outcomes of the study should be clearly stated in the methodology.
Response 9. Primary and secondary outcomes were clearly stated in the methodology section
- The sample size presented in this study is small. Because this is a prospective study, sample size calculation is mandatory. It is unclear how the authors calculated the sample size. How did they decide how many patients will be included in each study group? The methodology does not include a sample size calculation, which is one of the most important parameters for the study design. How do the authors know that the present sample size is sufficient to achieve statistical significance? This should be described in detail.
Response 10. Sample size is often calculated to collect data that when statistically analysed will produce conclusions strong enough to be generalized. In statistical terms, is calculated in order to have a good statistical power to conclusively test the original hypothesis. In these calculations, we need the significance level (normally defined at 0.05), the desired power level and the effect size, ie, expected differences among group. Because, our study is the first of its kind, there was no prior data or expectations on these differences (either if they exist or their magnitude). It was then difficult, or even impossible, to determine a good sample size. We therefore, collected all the data that we could in a period of 10 months and did the analysis that we now present.
- The quality of the English language is below the standard and should be significantly improved. The manuscript would benefit from editing by a native English speaker or a professional language editor to improve grammar and readability.
Response 11. A professional language editor, namely IJMS editor, will review the final text, after all the questions that the revisers have queried have been answered accordingly. If the revisers want it now, I will do it before ressubmission
Reviewer 2 Report
Overall remarks. The authors report "a prospective single-center study to evaluate" eosinophil biomarker "levels in appendicular lavage fluid (ALF) and ... serum of ... patients with acute phlegmonous ... appendicitis, ...acute gangrenous appendicitis, "vs ".. normal controls". The results indicate an intense local (ALF>serum) TH2 response in eosinophil biomarkers (eosinophil 20 cationic protein, ECP, and eosinophil peroxidase, EP) were measured in both acute phlegmonous and gangrenous appendicitis groups relative to control patient samples. These biomarkers were also significant in serum, though at lower levels than the local (lavage sourced) gut tissue response. Thus, both TH1 (prior reports from this and other groups) and TH2 responses appear active in acute appendicitis, consistent with this groups prior reports of Th2 cytokines (IL-4, IL-5 and IL-9) and IgE levels in appendicular lavage. The receiver operator characteristics curve in figure 5 and the correlations in Table 5 are especially supportive of the TH2 response and the potential diagnostic value in measuring ECP and EP in serum as a predictor of gut tissue eosinophil presence, even though blood eosinophil counts were not predictive of the serum or gut tissue lavage levels of eosinophil proteins (table 6). The match of eosinophil proteins to clinical events of peritonitis and gut perforation is also insightful, though ROC indicates blood ECP association with perforation is limited, though some sample size (10% of patients with perforation) may impact that specific association. The discussion is balanced, fair and consistent with the literature and the current data. The finding of eosinophil cationic proteins in ALF and peripheral blood and the association analysis with clinical events in these patients is novel.
Minor concerns:
1. p =0.000 is not helpful in tables 1-4 nor in text, such as lines 112-138 and beyond. Simply state the value limit, most seem to be only one decimal more than used in your current texts, tables and figure legends. Just a minor statistical inconsistency that p is not 0.
Author Response
We would like to thank the reviewers for their detailed comments and constructive suggestions to improve our manuscript. We have addressed all of the reviewer’s comments and concerns, hoping that our manuscript is now suitable for publication in this journal.
- p =0.000 is not helpful in tables 1-4 nor in text, such as lines 112-138 and beyond. Simply state the value limit, most seem to be only one decimal more than used in your current texts, tables and figure legends. Just a minor statistical inconsistency that p is not 0.
Response 1. P=0-000 was eradicate from the text.
Round 2
Reviewer 1 Report
The authors adequately answered all questions raised by the peer review. The manuscript can be accepted in its present form, without further modifications.
The quality of the English language is below the standard and should be significantly improved. The manuscript would benefit from editing by a native English speaker or a professional language editor to improve grammar and readability.